# Digital Droplet PCR is a Specific and Sensitive Tool for Detecting IDH2 Mutations in Acute Myeloid LeuKemia Patients

**DOI:** 10.3390/cancers12071738

**Published:** 2020-06-30

**Authors:** Susanna Grassi, Francesca Guerrini, Elena Ciabatti, Riccardo Puccetti, Serena Salehzadeh, Maria Rita Metelli, Alessia Di Vita, Cristiana Domenichini, Francesco Caracciolo, Enrico Orciuolo, Matteo Pelosini, Elisa Mazzantini, Pietro Rossi, Francesco Mazziotta, Mario Petrini, Sara Galimberti

**Affiliations:** 1Department of Clinical and Experimental Medicine, University of Pisa, 56126 Pisa, Italy; guerrinifra@libero.it (F.G.); elenaciabatti@gmail.com (E.C.); r.puccetti1@studenti.unipi.it (R.P.); salehzadeh.serena@gmail.com (S.S.); elisa.mazzantini91@gmail.com (E.M.); rossietro@gmail.com (P.R.); francemazziotta@gmail.com (F.M.); mario.petrini@med.unipi.it (M.P.); sara.galimberti@med.unipi.it (S.G.); 2Hematology, Azienda Ospedaliero Universitaria Pisana (AOUP), 56126 Pisa, Italy; m.metelli@ao-pisa.toscana.it (M.R.M.); alessiadivita83@gmail.com (A.D.V.); cristiana.domenichini@tin.it (C.D.); francaracciolo@gmail.com (F.C.); e.orciuolo@alumni.sssup.it (E.O.); mpelo78@hotmail.com (M.P.); 3Department of Medical Biotechnologies, University of Siena, 53100 Siena, Italy

**Keywords:** digital PCR, IDH2, AML, MRD, CPX-351, Enasidenib

## Abstract

Isocitrate dehydrogenase 1 and 2 (IDH1 and IDH2) interfere with cellular metabolism contributing to oncogenesis. Mutations of IDH2 at R140 and R172 residues are observed in 20% of acute myeloid leukemias (AML), and the availability of the IDH2 inhibitor Enasidenib made IDH2 mutational screening a clinical need. The aim of this study was to set a new quantitative polymerase chain reaction (PCR) technique, the drop-off digital droplet PCR (drop-off ddPCR), as a sensitive and accurate tool for detecting IDH2 mutations. With this technique we tested 60 AML patients. Sanger sequencing identified 8/60 (13.5%) mutated cases, while ddPCR and the amplification refractory mutation system (ARMS) PCR, used as a reference technique, identified mutations in 13/60 (21.6%) cases. When the outcome of IDH2-mutated was compared to that of wild-type patients, no significant difference in terms of quality of response, overall survival, or progression-free survival was observed. Finally, we monitored IDH2 mutations during follow-up in nine cases, finding that IDH2 can be considered a valid marker of minimal residual disease (MRD) in 2/3 of our patients. In conclusion, a rapid screening of IDH2 mutations is now a clinical need well satisfied by ddPCR, but the role of IDH2 as a marker for MRD still remains a matter of debate.

## 1. Introduction

Acquired somatic mutations in the isocitrate dehydrogenase 1 and 2 genes (IDH1 and IDH2), have been reported in acute myeloid leukemia (AML) [1], myelodysplastic syndromes (MDS), and chronic myeloproliferative neoplasms (MPN) [2,3]. They occur into conserved active sites and mutant enzymes cannot still catalyze the oxidative decarboxylation. As a result of such block of conversion of isocitrate to alpha-ketoglutarate in the Krebs cycle, there are increased levels of the oncometabolite 2-hydroxyglutarate [4]. This accumulation promotes oncogenic effects and inhibits normal cell differentiation and enzymatic activities, such as the TET2-dependent DNA hydroxymethylation, histone demethylation, and HIF-1α activation [5].

In particular, IDH2 mutations are found in about 20% of de novo AMLs, mainly in patients with normal karyotype. They are single-nucleotide variants involving the exon 4 at the arginine hotspot R140 or R172. Today, the screening of IDH1 and IDH2 mutations play an important role in the classification of AML and for setting ab initio as the most effective treatment, even due to the recent availability of IDH1 and IDH2 inhibitors (Ivosidenib and Enasidenib). In a phase 1/2 clinical trial enrolling relapsed/refractory AML patients, Enasidenib offered 40% of the overall response rate (ORR), 19.3% of complete remissions (CRs), with a median overall survival (OS) of 19.7 months in patients achieving CR [6]. Regarding the role of IDH2 mutations as a valid marker of minimal residual disease (MRD), this topic is still a matter of debate. Indeed, the European guidelines do not include IDH in the list of genes to be considered as valid MRD biomarkers [7], but some authors reported a promising predictive role of IDH2 mutations in the MRD assessment when digital droplet PCR (ddPCR) is adopted [8]. 

For many years Sanger sequencing has been considered as the gold-standard method for detecting IDH2 mutations also, although it is not a quantitative method and has limited sensitivity (10–15%). Thus, alternative methods, such as high-resolution melting (HRM), allele-specific quantitative PCR based on the amplification refractory mutation system technique (ARMS PCR), ddPCR and next-generation sequencing (NGS), started to be implemented [9,10,11].

The ddPCR is based on the water-oil emulsion droplet technology that allows fractionating the template DNA into thousands of nanoliter-sized droplets where the amplification occurs. The ddPCR technology, similar to real-time PCR, uses TaqMan probe-based assays, even if results of the reaction are evaluated not in “real-time” but at the “end-point”. After amplification, each droplet is analyzed to assess the presence of a positive or negative signal, using the Poisson’s statistics to normalize and determine the target concentration in the original sample. The advantages of this technique are represented by the increased signal-to-noise ratio and by the absolute quantification without a need of standard curves. 

In this research, we developed a new drop-off ddPCR strategy that, differently from the “classical” ddPCR, is able to identify all possible nucleotide substitutions at the same codon in a single reaction. After setting the method, we compared it to the Sanger sequencing and the ARMS PCR (considered as “classical” gold-standard methods for mutational screening), showing a sensitivity of ddPCR comparable to that of ARMS PCR and higher than that of Sanger. Finally, we evaluated the possible role of IDH2 mutations in our AML cohort, in terms of quality of response to an induction/consolidation treatment and as a possible useful tool to monitor MRD. 

Our results confirm that the drop-off ddPCR is a rapid, sensitive, quantitative, and cost-effective method for IDH2 genotyping to be carried out in the diagnostic hematological routine.

## 2. Results

### 2.1. Drop-Off ddPCR Setting and Validation

The first block of experiments has been performed in order to set the optimal amplification conditions and all parameters that could allow the best separation between mutated and wild-type (WT) droplet clouds (therefore, avoiding the undesirable “rain” effect). In this phase, we evaluated different experimental conditions, such as increasing DNA quantities (from 16 to 80 ng), different annealing temperatures (from 54 to 58 °C), and variable primers/probes concentrations. The best results were obtained when we started with 16 ng of DNA. The mixture for detection of R140 mutations contained 1.1 µL of the primers/probe belonging to the reference sequence and 1.1 µL to the WT one, in a final reaction volume of 22 µL. The ddPCR for the R172 assay contained 1.38 µL of the reference primers/probe and 0.82 µL of the WT, in the same final PCR volume. The amplification cycling conditions were: 95 °C (2.0 °C/s ramp) for 10 min (cycle 1), then 94 °C (2.0 °C/s ramp) for 30 s, 56 °C for 1 min (cycle 2–40), and 98 °C (2.0 °C/s ramp) for 10 min (final cycle). 

Once the tests were set, we evaluated their specificity and sensitivity. The specificity was determined adopting the “GeneArt Strings DNA Fragments” (ThermoFisher, Waltham, MA, USA), double strand DNA (dsDNA) linear fragments that we used as positive controls for different point mutations (R140W, R140L, R140Q, R140G and R172W, R172G, R172S, R172M, R172K). As expected, amplification products were obtained for all mutated sequences. 

Finally, we evaluated the sensitivity of our method on a dilution series containing different quantities of the mutated controls mixed with IDH2 WT DNA. All tested mutations were detected up to 0.1% dilution (1x10^-3^).

### 2.2. Assessment of IDH2 Mutations and Comparison among ddPCR, ARMS PCR, and Sanger

After setting the method, we analyzed by the drop-off ddPCR 60 bone marrow samples harvested at diagnosis whose IDH2 mutational status had been previously assessed by Sanger sequencing. By Sanger, eight cases (13.3%) resulted mutated, six carrying the R140Q (c.419G>A) and two the R172K (c.515G>A) mutation. In each mutated case, a single arginine point mutation was detected, indicating that IDH2 mutations are mutually exclusive. 

Since it is well known that Sanger is a low-sensitive technique, we decided to test our cases with the molecular PCR tool that is considered a sensitive reference technique for point mutations detection: The ARMS PCR. By this method, we detected IDH2 mutations in 13 out of 60 patients (21.7%), with the R140Q (c.419G>A) being the most frequent nucleotide change.

Finally, we screened the same DNAs by the drop-off ddPCR: This technique allowed us to identify mutations in 13/60 cases (21.7%), perfectly in line with results coming from ARMS PCR, therefore, recovering five cases (8.3% of the whole series) compared to Sanger sequencing (Table 1). 

In mutated patients, the median IDH2 allele burden was 13%, with an inter-patients very wide range (0.39–43.90%). 

When we analyzed the results performing the Cohen’s kappa coefficient (κ) test, we found that there was a “perfect” concordance between the ddPCR and ARMS PCR and a “good” concordance between the ddPCR and Sanger (Table 2).

Then, in order to try to better understand why five cases resulted in wild-type by Sanger but mutated by ddPCR, we focused on their mutational burden: As expected, these discordant cases (four with R140Q and one with the R172K mutation) showed low mutational levels, ranging from 0.39% to 12%. Since the sensitivity of Sanger is around 15%, it is clear why these “subclonal” mutations had not been detected by Sanger sequencing.

### 2.3. Clinical Impact of IDH2 Mutations

At the beginning, we checked if there was a correlation between IDH2 mutations and any of the clinical features, such as sex, age, median white blood cells count (WBC) hemoglobin (Hb) or platelets (PLT) count, the World Health Organization (WHO) classification, concomitant presence of any different mutations, and the European Leukemia Network (ELN) risk. As shown in Table 3, no significant differences between the 13 IDH2-mutated and the 47 unmutated cases have been observed. 

As an induction regimen, two patients received decitabine and two CPX-351 (a liposomal form of daunorubicine and aracytin combination); the remaining ones received daunorubicine and aracytin (according to the 7 + 3 or 5 + 2 schemes). Overall, 16 patients underwent allogeneic stem cell transplantation (alloSCT), 10 in the IDH2-mutated and six in the wild-type cohort. Five of the 13 IDH2-mutated patients received the IDH2 inhibitor Enasidenib at relapse.

When we assessed if IDH2 mutations might condition the quality of response after induction/consolidation treatment, we found that there was not a statistically significant difference, either in terms of the overall response rate (ORR) (69% in IDH2-WT vs. 84% in the IDH2-mutated subgroup) or CR rate (53% in IDH2-WT vs. 68% in the IDH2-mutated subgroup) (*p* = 0.47). Interestingly, a patient receiving CPX-351 reached immediately a complete MRD-negative response just after the first induction cycle, an optimal response that allowed him to rapidly undergo to a successful allogeneic stem cell transplantation (alloSCT).

Then, we compared the overall survival (OS) and progression-free survival (PFS) in the two cohorts: 14 patients (23%) died from disease progression in the first three months from diagnosis; the median OS of the entire series was 12.2 months, longer for patients <65 years and for those who received alloSCT (median OS of younger/transplanted cases, 22 months, *p* = 0.007). 

As shown in Figure 1, OS was not significantly influenced by the IDH2 mutational status: Indeed, the three-year OS was 34% for IDH2-unmutated vs. 30% for the IDH2-mutated subjects (*p* = ns).

The median PFS of the entire cohort was 10 months, with a three-year PFS of 22%. Even in this case, the outcome was longer for transplanted cases (median PFS: 14.6 months, *p* = 0.012). As shown in Figure 2, also the PFS length was not significantly influenced by the IDH2 mutational status: Indeed, the three-year PFS was 23% for IDH2-unmutated versus 19% for the IDH2-mutated subjects (*p* = ns). 

In conclusion, in our series IDH2 mutations did not predict a long-term outcome in all analyzed subgroups.

Another interesting finding of our study concerns the association between IDH2 and additional mutations: At diagnosis, 32% of the overall series and 69% of IDH2-mutated patients presented at least one concomitant further mutation (see Table 4). IDH2 was combined with FLT3-ITD or CBF-MYH16 mutations in one case, respectively, with RUNX1/RUNX1T1 or c-KIT in two cases, with NPM1 in three patients and with N-RAS in five cases, while only one patient was mutated also for IDH1. Two patients presented overall four different mutations; differently from that we observed in our previous experience, where the presence of additional mutations significantly impaired the outcome of AML patients [11], in the present series we did not observe a different outcome of patients with or without additional mutations, neither in terms of PFS or OS, probably for the low number of cases.

When we considered the outcome of the five patients treated with Enasidenib compared to that of IDH2-mutated patients treated only with “conventional” re-treatment, the use of the IDH2 inhibitor gave an advantage, with a median OS from the diagnosis of 40 months for cases receiving the “target therapy” versus 10 months of the remaining ones (*p* = 0.05). Obviously, the number of cases is not significant, but this result might sustain also in the “real world” the use of Enasidenib in IDH2-mutated patients.

Finally, we used IDH2 mutations for assessing MRD in 37 samples from nine patients (in four cases residual DNA was not sufficient to perform further molecular tests). Figure 3, Figure 4 and Figure 5 show the behavior of IDH2 mutations during follow-up; cases #E and #F have been also tested for Wilms’ tumour 1 (WT1) gene expression and NPM1 mutational burden and case #I both for WT1 expression and FLT3 mutations. In six cases (#A, #B, #C, #D, #E, #F) IDH2 was a good marker of disease status; indeed, in cases #A, #B, #C, #D, the IDH2 mutational load well correlated with CR (Figure 3), and in cases #E, #F, the increased levels of IDH2 mutations predicted disease progression (Figure 4). On the contrary, in other three cases (#G, #H, #I), the behavior of IDH2 mutations did not correlate with the clinic outcome. In case #I, analogously to IDH2, the FLT3 mutation level decreased even if the patient was not responsive to therapy, while the precocious WT1 increased level predicted the poor patient’s outcome (Figure 5).

## 3. Discussion

This study for the first time presents a new molecular tool useful for detecting IDH2 mutations in AML patients: The drop-off ddPCR. Indeed, the recent availability of Enasidenib made the detection of IDH2 mutations a new clinical need. In a phase-I study, this inhibitor offered relapsed/refractory patients an overall response rate of 40.3%, 19.3% of CRs, and a median OS of 19.7 months, with a good tolerability profile (grade 3–4 indirect hyperbilirubinemia in 12% and differentiation syndrome in 7% of cases) [12]. These results have also been confirmed in the phase-III trial, even if with a lower median OS (9.3 months) [13]. In our study, in line with data from the literature, we found 21.6% of IDH2-mutated patients; the clinical characteristics of this cohort did not differ from those of the unmutated one. 

Interestingly, patients who received Enasidenib showed a prolonged median OS (40 vs. 10 months). Obviously, the number of mutated patients receiving this treatment is not enough to derive definitive conclusions about its efficacy and safety, but supports its use in the clinical practice, perhaps in association with chemotherapy or as consolidation/maintenance after achieving partial response or after allogeneic stem cell transplantation. 

About ddPCR, two other groups already proposed it as a good technique for IDH2 mutations detection: In 2018, Petrova et al. used this method for screening 90 AML patients: IDH1/IDH2 mutations were detected in 22 cases (24%), with only one case carrying both IDH1 and IDH2 mutations [8]. In particular, IDH2 mutations showed a prevalence of 14.4%, a percentage comparable with our results (21% of mutated cases). The authors used ddPCR for assessing MRD in 22 patients, observing a reduction of mutational burden in all cases except for two. Moreover, in four cases, IDH mutations well correlated with other “classical markers” of MRD: Two patients carried concomitant NPM1 mutations, one MLL/MTD rearrangement, and the other one CEBPa and DNMT3A mutations. The authors reported that IDH as a marker of MRD well correlated with the behavior of NPM1 mutations but not with mutations of CEBPa or DNMT3A. The authors also sustained that ddPCR was very sensitive, more than NGS [8]: Indeed, NGS has been previously proposed as a technique used for detecting IDH1/2 mutations [14], but its sensitivity of 1% appears significantly lower than that offered by the new ddPCR. However, NGS, by massive sequencing, might be more informative than the PCR methods and might detect an eventual clonal evolution that nucleotide-specific PCR assays are not able to see (because they are set for each specific nucleotide aberration). Recently, the group from Turin compared a further new molecular PCR method, the Peptide Nucleic Acid (PNA) PCR clamping, to ddPCR and Sanger sequencing. At diagnosis, ddPCR identified as mutated 16.2% of cases, PNA-PCR 14.8%, and Sanger 12.1%. This difference might be probably explained by the different sensitivity of the methods (1x10^−3^ for ddPCR, 1% for PNA-PCR clamping, and 10–15% for Sanger) [15].

In the present study, we developed a new type of ddPCR, the drop-off ddPCR, that represents a cheap, accurate, and sensitive method for screening IDH2 mutations in AML. Indeed, the goal of this new technique is the possibility to combine the absolute quantification and the detection of potentially all mutations present in a hot spot target region. In this way, we can overcome some of the disadvantages presented by the real-time PCR or PNA-PCR clamping, that allow to recognize only nucleotide-specific changes, therefore, requiring a specific test for each of the IDH2 possible mutations, with consequent higher costs and longer times to produce results. 

Contrariwise, the drop-off ddPCR identifies any mutation in a unique reaction, being cheaper and faster. Consequently, this technique could represent the most valid approach for screening patients for IDH2 mutations and immediately starting the most effective treatment. In addition to Enasidenib, the use of CPX-351 could be a very favourable one. At the ASH 2018, the results from the Italian compassionate use of CPX-351 have been reported: CR was observed in 86.3% of the 75 enrolled patients, with 45% of them who achieved also the MRD-negativity after the first cycle [16]. Interestingly, during the same meeting other groups reported that patients receiving CPX-351 achieved a 100% of response if IDH2-mutated [17], and Chiche et al. assessed that the favorable outcome was observed in patients after HSCT and CPX-351 erased the poor prognosis associated with unfavorable mutations [18], thus making CPX-351 an attracting therapeutic chance for this subgroup of AML patients. 

In our series, the patient who received this liposomal drug achieved an MRD-negative CR after only one cycle and rapidly went to a successful alloSCT. This observation further confirms that detection of IDH2 mutations is now fundamental for adopting ab initio as the best therapeutic strategy. In a phase I/II study, Enasidenib, used in combination with Azacytidine in IDH2-mutated patients not candidates for intensive chemotherapy, offered 68% of responses, 50% of CRs, and a longer duration of response. Interestingly, the combined treatment cleared the mutational burden more than azacytidine alone (median reduction: 69.3% vs. 14.1%), therefore, showing the benefit that Enasidenib may offer to AML patients [19]. 

Regarding the role of IDH2 mutations as a marker of MRD, a definitive conclusion may be not treated. With sensitivity of 0.1%, in our series, ddPCR was a good marker of MRD only in selected (6/9) patients: In a case that we monitored after allogeneic stem cell transplantation, only the IDH2 increase was predictive of relapse, whilst chimerism remained as a full donor and WT1 expression levels remained normal for two further months. 

Nevertheless, the potential of using IDH mutations as markers for MRD is still a matter of debate; indeed, ELN guidelines do not suggest the use of IDH2 as a tool for MRD investigation, while some authors sustained that IDH2 might be a good biomarker [8,15]. Perhaps, the predictive/prognostic role of IDH2 mutations may be influenced either by the clinical context (for example target versus conventional therapies) or by some pathogenetic aspects, such as concomitant mutations that can represent an indirect sign of higher genomic instability or clonal branching [20,21,22]. In our limited experience, the IDH2 mutations behavior was similar to that observed from NPM1 and FLT3 mutations, which prompts us to hypothesize that all mutations might be present in the same leukemic clone. At ASH 2019 it was reported that in relapsed AML cases the median number of coexistent mutations was three and that the most common mutations coexistent with IDH2 included NPM1 (7%), RUNX1 (6%), and FLT3 (5%) [23]. In our study, we observed that 32% of patients at diagnosis present at least two mutations and that IDH2 was associated with FLT3, RUNX1/RUNX1T1, NPM1, c-KIT, and N-RAS mutations (7–13% of cases). It could be also interesting to have the clonal evolution of IDH2 mutations under treatment, but we have monitored patients following ELN guidelines, and therefore, we have no data on the clonal evolution of mutations found. We could only observe that at diagnosis, the presence of several coexistent mutations did not significantly influence the outcome of our patients, this point must to be considered in larger studies. 

## 4. Materials and Methods 

### 4.1. Patients

This is a single-center, retrospective study; all patients enrolled in the study gave their informed consent to donate the leftovers of diagnostic samples for further non-profit scientific purposes. This consent had been approved by the Ethical Committee of the North-West Tuscany (CEAVNO) of the Azienda Ospedaliera Universitaria Pisana (AOUP). The enrolled patients were chosen among those treated at the Hematology Ward of Pisa (Italy) from January 2015 to December 2018, only on the basis of availability of residual stored DNA and adequate clinical follow-up. 

Sixty patients were tested, 33 men and 27 women, with a median age of 55.5 years (range, 19–79 years) (patients characteristics are listed in Table 5). In nine cases, the ddPCR was also employed for monitoring MRD, for a total of 84 mutational assessments. According to the international guidelines, at baseline all samples were tested for BCR/ABL1, RUNX1/RUNX1T1, CBFB/MYH11, and PML/RARa rearrangements, according to the Biomed 1 protocol [24], and for WT1 expression levels (by Ipsogen WT1^®^ ProfileQuant Kit, Qiagen, Hilden, Germany)). Moreover, mutational analyses of FLT3 and NPM1 were performed, as previously reported [25,26]. The risk categories were defined according to recommendations previously edited by the ELN [27].

### 4.2. Methods

#### 4.2.1. DNA Extraction

Genomic DNA was extracted from 350 µL of bone marrow or peripheral blood anticoagulated with EDTA using the automatic apparatus BioRobot EZ1 Advanced XL automated instrument (Qiagen, Hilden, Germany), according to the manufacturer’s instructions. The extracted DNA was then quantitated using the Thermo Scientific NanoDrop 2000 spectrophotometer (Thermo Fisher Scientific R, Wilmington, DE, USA). 

#### 4.2.2. Sanger Sequencing of IDH2 R140 and R172 Mutations

Direct Sanger sequencing was done by capillary electrophoresis using the ABI Prism BigDye Terminator Cycle Sequencing Kit 1.1 and Applied biosystem 3500 Genetic Analyzer (Waltham, MA, USA) after IDH2 DNA amplification by PCR. The primers flanking the target region and the reaction conditions have been performed as previously described by Marcucci et al. [28]. The resulting sequences have been finally compared to the wild-type IDH2 cDNA (GenBank Accession numbers NM_005896.2). 

#### 4.2.3. IDH2 R140 and R172 Mutations and Real-Time ARMS PCR

Molecular analysis of the IDH2 Q140 mutations was performed by using the qBiomarker Somatic Mutation PCR Arrays (Qiagen, Milan, Italy). Each array included hot spot mutations for ASXL1, TET2, IDH1, IDH2, N-RAS, WT1, c-KIT, RUNX1, DNMT3A, FLT3, and NPM1, for a total of 83 possible mutation sites. For IDH2, the point mutations represented in the PCR plate are listed in Table 6.

Each analysis was performed by QT-PCR using the ARMS technology, which is based on the discrimination between matched and mismatched primers specifically designed to amplify the mutated target DNA. The QT-PCR was performed on the CFX machine (BioRad R, Hercules, CA, USA). All samples and controls were amplified and analyzed according to the manufacturer’s instructions. The data analysis was based on the “ ΔCt” method: The amount of mutant DNA was calculated by the formula: “ΔCt sample” = Ct of mutated allele – Ct of wild-type copy number; “ΔCt healthy donor” = Ct of mutated allele – Ct of wild-type copy number; “ΔΔCt” = ΔCt sample – ΔCt healthy donor. A mutation call can be made when ΔΔCt was > 4; when ΔΔCt was < 3 the sample was considered as wild-type; when ΔΔCt was between 3 and 4, the specimen resulted borderline. Moreover, samples were defined as WT if Ct of the mutated allele was > 37 (total cycles of amplification = 40).

#### 4.2.4. IDH2 R140 and R172 Analysis by the Drop-Off ddPCR

The ddPCR is an emerging methodology that allows both screening and mutation monitoring. Many mutational assays are currently available to detect single nucleotide mutations. These tests involve the use of a pair of primers and two probes conjugated with different fluorochromes (FAM or HEX), the former being specific for the mutated allele and the latter one for the WT allele. 

Since several variants can be detected in the IDH2 target sites, we needed to create an assay that could allow us to detect any mutations occurring in a specific nucleotide site. For this reason, we decided to use the “drop-off” strategy. A drop-off ddPCR FAM/HEX Assay (Biorad^®^) method was used; it requires a single pair of probes to detect and quantify different mutations of the target hot spot region in a single reaction. The FAM-labeled probe binds a reference sequence distant from the target but still within the same amplicon, while the HEX probe binds the wild-type sequence precisely in the target site only. In a 2D-plot, wild-type samples show signals from both FAM (reference sequence) and HEX probes (WT), while the mutated ones display only the FAM signal (Figure 6). This method has been developed both for R140 and for R172 target regions. Primers and probes are listed in Table 7.

The ddPCR was performed using the QX200 platform (BioRad R, Hercules, CA, USA), consisting of two instruments: The droplet generator and the droplet reader. The droplet generator divides the sample by creating about 20,000 partitions (droplets). The droplets are then transferred into PCR plates and, at the end of the amplification cycles, placed into the droplet reader. Here, each droplet is detected as mutated or wild-type by issuing specific fluorescence signals (FAM for the mutation and FAM-HEX for the wild-type). These signals, after being counted, were analyzed and redistributed according to the Poisson’s algorithm.

#### 4.2.5. Statistical Analysis 

The relationships between IDH2 mutations and patients’ characteristics were calculated by the SPSS 22.0 software (SPSS Inc, Bologna, Italy). Categorical data was described by absolute frequency, and continuous data by median and interquartile range. To compare the qualitative and the quantitative variables, the Kruskal-Wallis test followed by comparisons with the Bonferroni’s inequality or Mann-Whitney tests were used. OS was calculated from the date of diagnosis to death or last follow-up; PFS was measured from the start of the induction to the last follow-up, disease progression, discontinuation of treatment or relapse. The Kaplan-Meier method and the log-rank test were utilized to estimate the distribution of survival curves. For all analyses, a *p*-value ≤ 0.05 was considered as statistically significant. 

## 5. Conclusions

To sum up, our study for the first time has been able to give evidence that the drop-off ddPCR is a valid new molecular tool for detecting IDH2 mutations in AML patients. Its role during the follow-up still remains a matter of debate, and probably also as a marker for MRD in at least half of the cases, thus representing useful help for a correct and successful management of AML patients. Further studies, based on larger cohorts of patients, prospective instead of retrospective, are now required to confirm these findings and to help physicians to design patient-tailored valid therapeutic strategies.

## Figures and Tables

**Figure 1 cancers-12-01738-f001:**
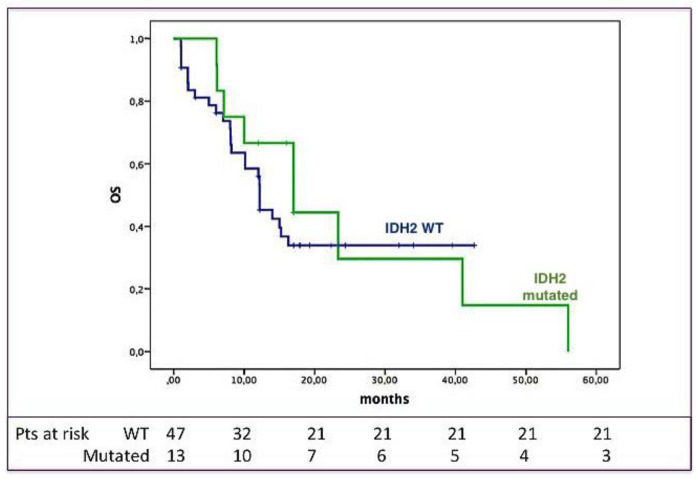
IDH2 mutational status and overall survival (OS). OS according to the presence/absence of IDH2 mutations; as shown, OS was not different between IDH2-unmutated (blue) and -mutated (green) patients.

**Figure 2 cancers-12-01738-f002:**
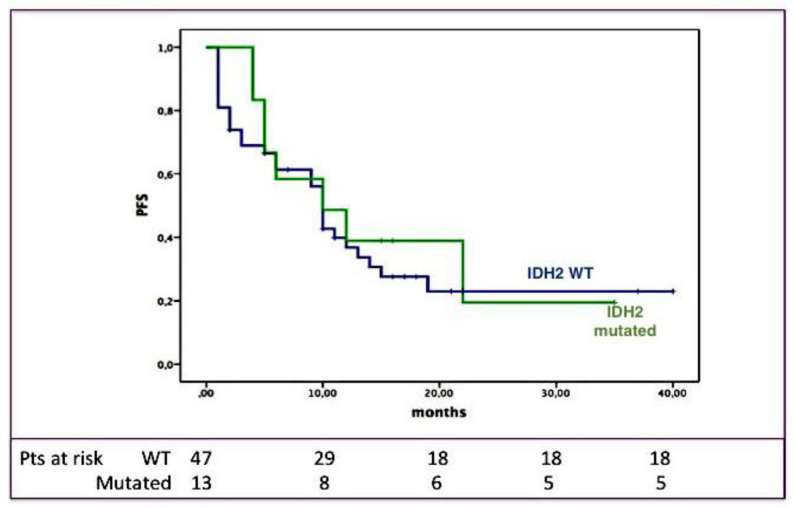
IDH2 mutational status and progression-free survival (PFS). PFS according to the presence/absence of IDH2 mutations; as shown, PFS was not different between IDH2-unmutated (blue) and -mutated (green) patients.

**Figure 3 cancers-12-01738-f003:**
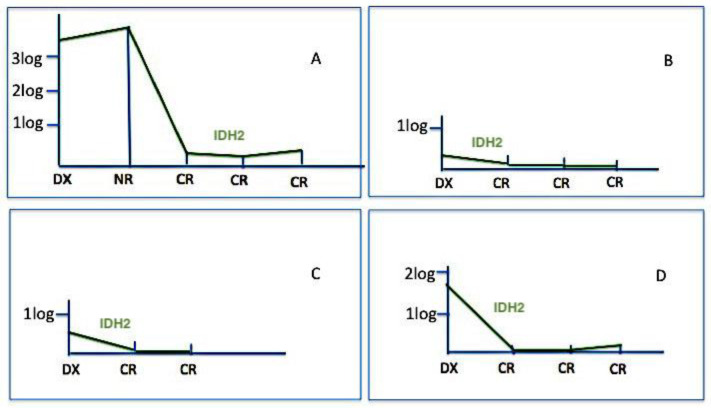
IDH2 mutation monitoring in complete remission (CR). In these four cases (#A, #B, #C, #D), IDH2 mutations well correlated with the clinical response; in these patients IDH2 might be considered as a good marker for minimal residual disease (MRD).

**Figure 4 cancers-12-01738-f004:**
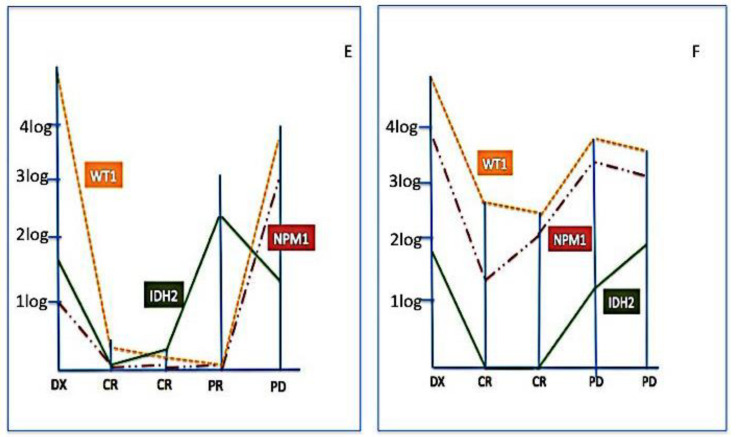
IDH2 mutation monitoring in disease progression (PD). These represent two cases (#E and #F) where MRD has been monitored by Wilms’ tumour 1 (WT1) gene expression (orange line), Nucleophosmin 1 (NPM1) mutational burden (red line), and IDH2 (green line). In both cases, all three markers (IDH2 = green line) presented the same trend, predicting the loss of response. Moreover, in these patients, IDH2 might be considered as a good marker for MRD.

**Figure 5 cancers-12-01738-f005:**
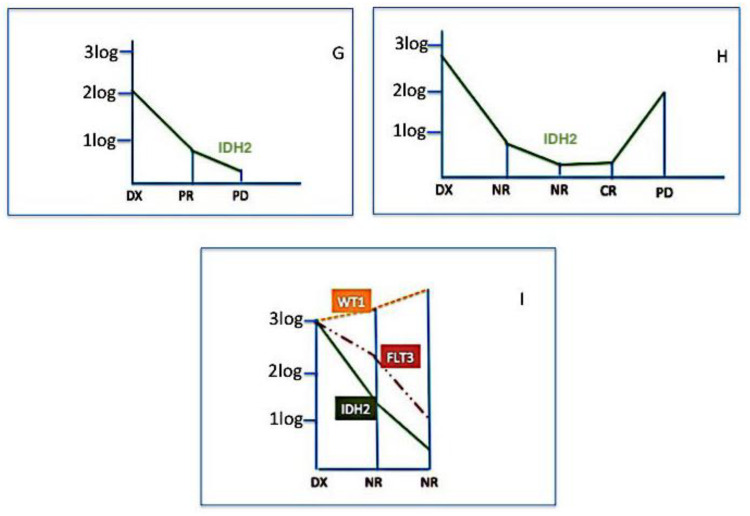
IDH2 mutation is not related to the clinical outcome. Three cases (#G, #H, #I) are represented in which a correlation between the IDH2 mutational status (green line) and clinical outcome has not been observed. Patient #I has been.also tested for WT1 expression (orange line) and FLT3 mutation (red line); only WT1 gene expression levels predicted the clinical outcome.

**Figure 6 cancers-12-01738-f006:**
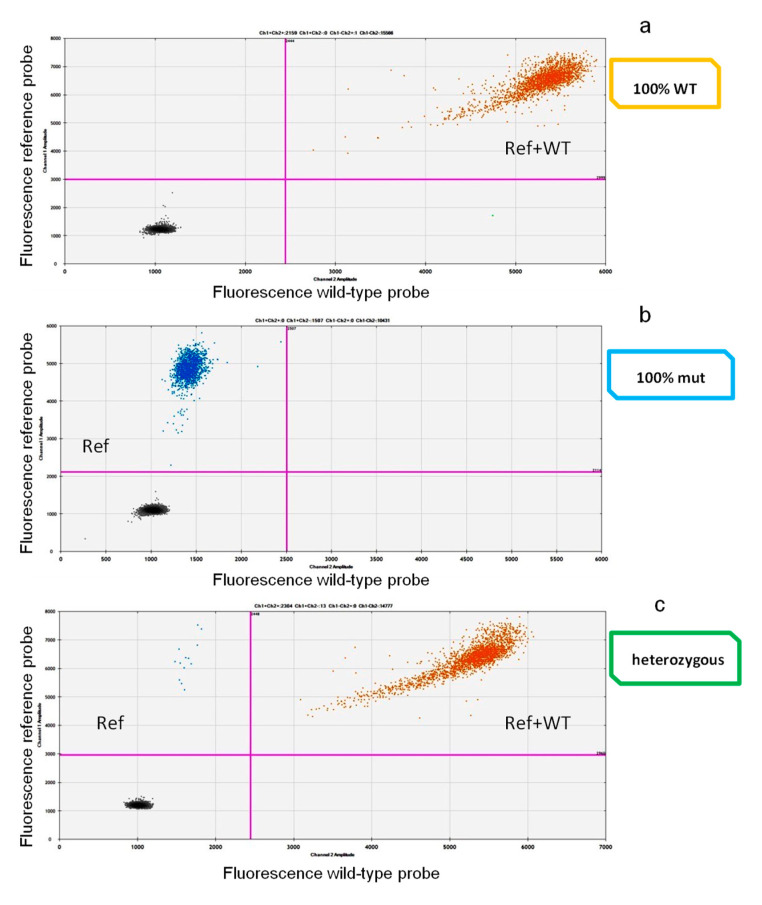
Two-dimensional (2D)-plot drop-off ddPCR. In the 2D plot, samples with different IDH2 genotypes are represented, with channel 1 fluorescence (reference probe) plotted against channel 2 fluorescence (WT probe). The droplets are arranged according to the fluorescence levels. In (**a**), a wild-type (WT) sample represented by a “double positive” population (in orange; reference and wild-type probe in the same droplet) (ref + wt). In (**b**), a 100% IDH2-mutated case, where only the reference probe (blue = ref) matched with the IDH2 sequence. In (**c**), a sample carrying the mutation in heterozygosity. This panel represents two droplets’ populations, the (with few events) mutated one (blue) (ref) and the (with a higher number of events) double positive (orange) (ref + wt) one.

**Table 1 cancers-12-01738-t001:** Isocitrate dehydrogenase 2 (IDH2) mutational status results.

		Sanger	ARMS PCR	Drop-Off ddPCR
Positive	R140Q	6	10	10
	R172K	2	3	3
	TOTAL (%)	8 (13.3%)	13 (21.7%)	13 (21.7%)
Negative		52 (86.7%)	47 (78.3%)	47 (78.3%)

**Table 2 cancers-12-01738-t002:** Concordance of the methods using Cohen’s kappa coefficient. (Test A: Amplification refractory mutation system polymerase chain reaction (ARMS PCR) and drop-off digital PCR (ddPCR); Test B: Sanger sequencing).

		Test A		
		+	**–**	Total
**Test B**	+	8	0	8
	-	5	47	52
	Total	13	47	60
Concordance = 0.917				
K = 0.715				

**Table 3 cancers-12-01738-t003:** Comparison of clinical characteristics between IDH2-mutated and -unmutated groups.

Clinical Features	IDH2-WT n. (%)	IDH2-MUTATED n. (%)	*p*
**SEX**			
M	25 (53%)	7 (54%)	ns
F	22 (47%)	6 (46%)	
**AGE**			
**< 65y**	31 (66%)	9 (69%)	ns
**≥65 y**	16 (34%)	4 (31%)	
**WHO classification**			
**AML with recurrent abnormalities**	17 (37%)	4 (31%)	ns
**Post-myelodysplasia**	6 (12%)	3 (23%)	
**Therapy-related**	3 (6%)	0 (0%)	
**NOS**	21 (45%)	6 (46%)	
**CYTOGENETIC score**			
**Good**	8 (17%)	3 (23%)	ns
**Intermediate**	22 (47%)	7 (54%)	
**Poor**	17 (36%)	3 (23%)	
**ELN score**			
**Good**	4 (9%)	1 (8%)	ns
**Intermediate**	27 (57%)	6 (46%)	
**Poor**	16 (34%)	6 (46%)	
**RAS mutations**			
No	36 (77%)	11 (85%)	ns
Yes	11 (23%)	2 (15%)	
**c-KIT mutations**			
No	42 (89%)	12 (92%)	ns
Yes	5 (11%)	1 (8%)	
FLT3 mutations			
No	36 (77%)	11 (85%)	ns
Yes	11 (23%)	2 (15%)	
NPM1 mutations			
No	37 (78%)	11 (85%)	ns
Yes	10 (22%)	2 (15%)	
CBF mutations			
No	40 (85%)	13 (100%)	ns
Yes	7 (15%)	0 (0%)	
WBC, median, × 109/L	21.8	20.1	ns
Hb, median, g/dL	10.2	10.4	ns
PLT, median, × 109/L	937	756	ns
Blasts %, median	40	34	ns

**Table 4 cancers-12-01738-t004:** Number and type of co-mutations for all IDH2-mutated patients.

IDH2 Positive Pts		Additional Mutations								
			NPM1	Ckit	FLT3-ITD	INV16	Runx1	IDH1	NRas	None
1	R140Q				+					
2	R140Q		+				+		+	
3	R140Q						+	+	+	
4	R172K			+						
5	R140Q		+						+	
6	R172K			+						
7	R140Q		+						+	
8	R140Q									+
9	R140Q									+
10	R140Q									+
11	R140Q									+
12	R172K					+				
13	R140Q								+	
		n°pts (%)	3 (23%)	2 (15%)	1 (7%)	1 (7%)	2 (15%)	1 (7%)	5 (38%)	4 (31%)

**Table 5 cancers-12-01738-t005:** Clinical features of 60 patients enrolled in the study.

Clinical Features	PTS N. (%)
**Sex**	
M	32 (53%)
F	28 (47%)
**Age**	
<65 y	40 (66%)
≥65 y	20 (34%)
**WHO classification**	
AML with recurrent abnormalities	21 (35%)
Post-myelodysplasia	9 (15%)
Therapy-related	3 (5%)
NOS	27 (45%)
**Cytogenetic score**	
Good	11 (18%)
Intermediate	29 (48%)
Poor	20 (34%)
**ELN score**	
Good	5 (8%)
Intermediate	33 (55%)
Poor	22 (36%)
WBC, median (range) × 10^9^/L	6.05 (1–118)
Hb, median (range) g/dL	9.7 (5–13)
PLT, median (range) × 10^9^/L	59.5 (10–344)
Blasts %, median (range)	45 (20–85%)
**RAS mutations**	
No	47 (78%)
Yes	13 (22%)
**c-KIT mutations**	
No	54 (90%)
Yes	6 (10%)
FLT3 mutations	
No	47 (78%)
Yes	13 (22%)
NPM1 mutations	
No	48 (80%)
Yes	12 (20%)
CBF mutations	
No	53 (88%)
Yes	7 (12%)

**Table 6 cancers-12-01738-t006:** IDH2 point mutations in qBiomarker Somatic Mutation PCR Arrays.

Gene	COSMIC ID	Nucleotide Change	Acid Change	Assay Catalog
IDH2	41877	c.418C>T	R140W	SMPH025023A
IDH2	41590	c.419G>A	R140Q	SMPH025021A
IDH2	41875	c.419G>T	R140L	SMPH025022A
IDH2	34039	c.514A>T	R172W	SMPH006598A
IDH2	33733	c.515G>A	R172K	SMPH006597A

**Table 7 cancers-12-01738-t007:** Drop off ddPCR primers and probes sequences.

	Primers	Probes
**IDH2 R140 REFERENCE**	Fwd [9 µM] GATGGGCTTGGTCCAG Rvs [9 µM] AAGAAGATGTGGAAAAGTCC	Probe [5 µM] 6-FAM ACATCCCACGCCTAGT
**IDH2 R140 WT**	Fwd [9 µM] GATGGGCTTGGTCCAG Rvs [9 µM] AAGAAGATGTGGAAAAGTCC	Probe [5 µM] HEX CCAGGATGTTCCGGATAG
**IDH2 R172 REFERENCE**	Fwd [9 µM] CTCCACCCTGGCCTA Rvs [9 µM] CTGGACCAAGCCCATC	Probe [5 µM] 6-FAM TCGCCATGGGCGTGCC
**IDH2 R172 WT**	Fws [9 µM] CTCCACCCTGGCCTA Rvs [9 µM] CTGGACCAAGCCCATC	Probe [5 µM] HEX ATTGGCAGGCACGCC

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
