# Peer review of "Digital Droplet PCR is a Specific and Sensitive Tool for Detecting IDH2 Mutations in Acute Myeloid LeuKemia Patients"

_cancers, 2020, doi:10.3390/cancers12071738_

Round 1

Reviewer 1 Report

The paper by Grassi S et al. reports on the set up, validation and application of digital droplet PCR to study IDH2 mutations. Authors demonstrate that ddPCR relibly detects IDH2 mutations, showing a higher sensitivy than Sanger sequencing, thus allowing the identification of IDH2 mutations in 13% of AML cases. Although, IDH2 mutations are not prognosically relevant and cannot be used as stable molecular marker for disease monitoring, it predicts the sensibility to specific IDH inhibitors, such as enasidenib or ivosidenib, that have been demonstrated to improve the outcome of relapsed IDH-mutated AML.

It is worth mentioning that diverse types of newly available PCR tecnhologies, such as allele specific PCR, have been demonstrated to be more sensitive than Sanger Seq in detecting IDH2 mutations.

English must be heavily revised throughout the text

Minor

Acronyms must be explained the first time they are used (see ARMS PCR in the abstract)

ABSTRACT

Authors mentioned the ARMS PCR (please specify the acronym) which is not the "topic" of the paper. Was it used as control to validate ddPCR results? This should be specified.

They assessed that IDH2 is a valid marker for MRD monitoring; it is quite hard to arise at these conclusions as IDH2 appeared to correlate with the clinical outcome only in 66% of cases. Therefore, like FLT3 it cannot be considered a valid molecular marker for disease monitoring, except in selected cases.

INTRODUCTION

It is not necessary to mention the involvement of IDH genes in glioblastoma, since it is only one of tumours other than leukemias in which these genes can be involved

Line 6 of page 2: the word “today” is repeted twice in the same sentence

Lines 27-30 of page 2: delete the last part of the sentence “which allows us to……”too technical not nedeed in the introduction

RESULTS

2.2. Please re-write the sentence that explain IDH mutations are mutually exclusive. And also the sentence describing the mutational burden and range. They should be explained clearer.

The last sentence of this paraghraph does not clearly expose the differences between assessement of IDH2 or FLT3 mutation, or WT1 expression.

Reviewer 2 Report

Grassi et al. submitted to Cancers the manuscript entitled "digital droplet PCR is a specific and sensitive tool for detecting IDH2 mutations in acute myeloid leukemia patients". The purpose of the study is original and well written. Even if patient number is low, results supported conclusions.

Major points:

  1. It could be informative to have number and type of co mutations for all IDH2 mutated patients
  2. It could be also interesting to have the clonal evolution of IDH2 mutations under treatment and maybe discussed correlation or not with MRD

Minor points:

  1. Two abstracts (one italian and one french) presented at ASH meeting suggested a deeper response on CPX-351: I suggest to cite the both.
  2. You could discuss also the better sensibility of ddPCR but only a specificity on ONE mutation while NGS is less sensitive but more informative and could identify some clonal evolution.  

Round 2

Reviewer 1 Report

Authors have addressed  all requests

Author Response

Response to Reviewer 1 Comments

Point 1. Authors have addressed all requests

Response 1: Thank You very much for Your suggestions and further approval.

Reviewer 2 Report

The authors answered almost to all questions.

Minor comment: the authors cited italian abstract for CPX but the y have to cite also the french abstract (Chiche et al.) 
